# Persistence of Ebola virus in semen among Ebola virus disease survivors in Sierra Leone: A cohort study of frequency, duration, and risk factors

A. E. Thorson[1,2☺]*, G. F. Deen[3☺], K. T. Bernstein[4☺], W. J. Liu[5☺], F. Yamba[3], N. Habib[1], F. R. Sesay[6], P. Gaillard[1], T. A. Massaquoi[6], S. L. R. McDonald[1], Y. Zhang[5], K. N. Durski[7], S. Singaravelu[7], E. Ervin[4], H. Liu[5], A. Coursier[7], J. E. Marrinan[1], A. Ariyarajah[7], M. Carino[1], P. Formenty[7], U. Ströher[4], M. Lamunu[7], G. Wu[5], F. Sahr[6‡], W. Xu[5‡], B. Knust[4‡], N. Brouet[1‡], on behalf of the Sierra Leone Ebola Virus Persistence Study Group[¶]

**1** UNDP/UNFPA/UNICEF/WHO/World Bank Special Programme of Research, Development and Research Training in Human Reproduction, Department of Sexual and Reproductive Health and Research, World Health Organization, Geneva, Switzerland, **2** Department of Global Public Health, Karolinska Institutet, Stockholm, Sweden, **3** Sierra Leone Ministry of Health and Sanitation, Freetown, Sierra Leone, **4** Centers for Disease Control and Prevention, Atlanta, Georgia, United States of America, **5** National Institute for Viral Disease Control and Prevention, Chinese Center for Disease Control and Prevention, Beijing, China, **6** Sierra Leone Ministry of Defense, Freetown, Sierra Leone, **7** Department of Health Emergency Interventions, World Health Organization, Geneva, Switzerland

☺ These authors contributed equally to this work.
‡ These authors are joint senior authors on this work.
¶ Membership of the Sierra Leone Ebola Virus Persistence Study Group is provided in the Acknowledgments.
* thorsona@who.int

**Data Availability Statement:** Contractual agreements between the study parties (HRP/WHO, US CDC, China CDC, and the MoH SL) exists and

## Abstract

### Background

Sexual transmission chains of Ebola virus (EBOV) have been verified and linked to EBOV RNA persistence in semen, post-recovery. The rate of semen persistence over time, including the average duration of persistence among Ebola virus disease (EVD) survivors, is not well known. This cohort study aimed to analyze population estimates of EBOV RNA persistence rates in semen over time, and associated risk factors in a population of survivors from Sierra Leone.

### Methods and findings

In this cohort study from May 2015 to April 2017 in Sierra Leone, recruitment was conducted in 2 phases; the first enrolled 100 male participants from the Western Area District in the capital of Freetown, and the second enrolled 120 men from the Western Area District and from Lungi, Port Loko District. Mean age of participants was 31 years. The men provided semen for testing, analyzed by quantitative reverse transcription PCR (qRT-PCR) for the presence of EBOV RNA. Follow-up occurred every 2 weeks until the endpoint, defined as 2 consecutive negative qRT-PCR results of semen specimen testing for EBOV RNA. Participants were matched with the Sierra Leone EVD case database to retrieve cycle threshold (Ct) values from the qRT-PCR analysis done in blood during acute disease. A purposive

data rights reside with MOH-SL, the material owner. Inquiries related to the data, may be directed to Study Data Oversight Committee (SIS@who.int) with Subject: "Inquiry on Ebola Semen persistence study data".

**Funding:** NB received funding for this study from the Paul Allen family foundation. Award nr NA. https://pgafamilyfoundation.org/. The study team acknowledges the contributions of the WHO Ebola Response Program, the Paul G. Allen Family Foundation, and the UNDP (United Nations Development Program)–UNFPA (United Nations Population Fund)–UNICEF–WHO–World Bank Special Program of Research, Development and Research Training in Human Reproduction (HRP), a cosponsored program executed by the WHO; the US CDC, the China CDC, the Sierra Leone Ministry of Health and Sanitation and the Ministry of Defence, and the Joint United Nations Program on HIV/AIDS in support of the Sierra Leone Ebola Virus Persistence Study. NO - The PAF had no role in study design, data collection and analysis, decision to publish, or preparation of the manuscript. YES- individuals employed by the WHO and HRP have contributed to this work.

**Competing interests:** The authors have declared that no competing interests exist.

**Abbreviations:** Ct, cycle threshold; DRC, Democratic Republic of the Congo; EBOV, Ebola virus Zaire; ETU, Ebola treatment unit; EVD, Ebola virus disease; HR, hazard ratio; IC, interval-censored; qRT-PCR, quantitative reverse transcription PCR.

sampling strategy was used, and the included sample composition was compared to the national EVD survivor database to understand deviations from the general male survivor population. At 180 days (6 months) after Ebola treatment unit (ETU) discharge, the EBOV RNA semen positive rate was 75.4% (95% CI 66.9%–82.0%). The median persistence duration was 204 days, with 50% of men having cleared their semen of EBOV RNA after this time. At 270 days, persistence was 26.8% (95% CI 20.0%–34.2%), and at 360 days, 6.0% (95% CI 3.1%–10.2%). Longer persistence was significantly associated with severe acute disease, with probability of persistence in this population at 1 year at 10.1% (95% CI 4.6%–19.8%) compared to the probability approaching 0% for those with mild acute disease. Age showed a dose–response pattern, where the youngest men ($\leq$25 years) were 3.17 (95% CI 1.60, 6.29) times more likely to be EBOV RNA negative in semen, and men aged 26–35 years were 1.85 (95% CI 1.04, 3.28) times more likely to be negative, than men aged >35 years. Among participants with both severe acute EVD and a higher age (>35 years), persistence remained above 20% (95% CI 6.0%–50.6%) at 1 year. Uptake of safe sex recommendations 3 months after ETU discharge was low among a third of survivors. The sample was largely representative of male survivors in Sierra Leone. A limitation of this study is the lack of knowledge about infectiousness.

## Conclusions

In this study we observed that EBOV RNA persistence in semen was a frequent phenomenon, with high population rates over time. This finding will inform forthcoming updated recommendations on risk reduction strategies relating to sexual transmission of EBOV. Our findings support implementation of a semen testing program as part of epidemic preparedness and response. Further, the results will enable planning of the magnitude of testing and targeted counseling needs over time.

## Author summary

### Why was this study done?

- Evidence of traces of Ebola virus in semen had been reported among survivors for a very long time after their disease.

- This study originated out of a need to expand understanding beyond single reported cases, to gain knowledge on how many male survivors have Ebola virus persistence in semen over time, and for how long on average.

- We also aimed to analyze factors associated with longer persistence that could provide information on hypotheses on why some men carry virus in their semen for a very long time after the acute disease.

### What did the researchers do and find?

- In this study, 220 men who had survived Ebola disease in Sierra Leone during 2015–2016 provided semen specimens, and all of those who had traces of Ebola virus detected were followed every other week until their specimen turned negative.

- Seventy-five percent of these men still had traces of Ebola virus in their semen specimen at 6 months after being discharged following acute EVD, and 50% at 204 days.

- We also found that longer persistence of virus was significantly associated with severe acute EVD and older age.

## What do these findings mean?

- We show there is an urgent need to organize a national semen testing program as part of Ebola epidemic preparedness and response.

- The findings also show that in addition to testing, targeted safe sex counseling and free access to condoms should be a priority from the start of an outbreak.

- While we show that most of the survivors had traces of virus in semen at 6 months, more research is needed to understand the impact of viral RNA in semen on transmission of Ebola virus.

## Introduction

The 2014–2016, West African Ebola virus disease (EVD) epidemic was unprecedented in size, with 28,616 reported cases, 11,310 deaths [1], and socioeconomic consequences across Guinea, Liberia, and Sierra Leone [2,3]. The recent global public health emergency in the Democratic Republic of the Congo (DRC), with 3,481 Ebola cases reported at the end of the epidemic, further highlights the need for continued and strengthened efforts to combat EVD.

Clusters of EVD cases with sexual transmission being the most likely source of infection have been confirmed [4]. In Liberia, a male EVD survivor infected his sexual partner 179 days after his EVD onset [5,6]. Furthermore, genomic sequencing showed a man in Guinea infected his sexual partner 470 days after his EVD onset [7].

We performed a systematic review prior to study start, finding that evidence of Ebola virus (EBOV) persistence in survivors' body fluids and evidence of post-acute-disease sexual transmission used to be limited [8]. More recently, case reports from the West African epidemic suggest longer than expected persistence of EBOV ribonucleic acid (RNA) in semen, with detection in 1 male survivor's semen more than 2 years after Ebola treatment unit (ETU) discharge [9]. Most published reports on semen persistence so far share the limitations of not being designed to provide population inference of semen EBOV RNA persistence rates. Empirical evidence from longitudinal cohort studies with an adequate sample size and frequent and regular follow-up is missing, and the question of duration and population rate of semen persistence among male survivors is an important research gap. Schindell et al. [10] concluded their review on EBOV persistence in semen by stating that existing studies suggest a great variability in the duration of viral persistence. Similarly, very little is known of risk factors associated with longer persistence. Sissoko et al. [11] and Soka et al. [12] raise the hypothesis that men over 40 years have a higher risk of longer persistence, whereas the respective study designs limit full analytical proof of concept. Associations with other risk factors such as characteristics of acute disease or sociodemographics are unknown.

During the West African epidemic, in response to anecdotal reports of suspected sexual transmission, we implemented a prospective, longitudinal cohort study with comprehensive

follow-up of EVD survivors in Sierra Leone in 2015. The aim was to study duration and population rates of EBOV persistence in body fluids over time, and associations with risk factors. The cross-sectional baseline analysis was presented earlier [13]. Here we present a survival analysis of EBOV RNA semen persistence rates over time from the full cohort analysis, where participants were followed prospectively over time until specimens were negative for EBOV RNA. We also present a multivariate analysis of risk factors for persistence in semen.

## Methods

### Study design

This prospective, longitudinal cohort study included a convenience sample of 220 consenting male EVD survivors from Sierra Leone, aged 18 years or above, recruited at Freetown urban and Lungi semi-rural Ebola treatment sites between 27 May 2015 and 12 May 2016, and followed up until April 2017. Study participants were interviewed, and semen tested by quantitative reverse transcription PCR (qRT-PCR) for the presence of EBOV RNA at baseline [14]. Reaching study endpoint was defined as having 2 consecutive EBOV RNA negative specimens by qRT-PCR. Follow-up visits with interviews and semen testing occurred every 2 weeks, until reaching the endpoint. Methods for study implementation, data collection, and counseling, as well as baseline results, have been presented earlier [13–15]. Sample composition and representativity was analyzed in relation to the national EVD survivor database. Individual participant identifications were matched with the Sierra Leone EVD case and laboratory database to retrieve data on blood cycle threshold (Ct) values during acute disease.

### Ebola virus detection

Two different qRT-PCR assays were used for EBOV detection in this cohort: NP and VP40 target gene sequences [16–18] until 16 October 2015 at the US Centers for Disease Control and Prevention laboratory in Bo, Sierra Leone, then the GP and NP target gene sequences until study end at the Chinese Center for Disease Control and Prevention (China CDC) laboratory in Freetown using the Ebola Virus Real Time PCR Diagnostic Kit (Chinese FDA Registration Number: 20143402058; China CDC and Daan Gene, Guangzhou, China). The methods were evaluated and compared using the WHO EBOV proficiency test panel, and the sensitivity and specificity of the 2 methods were assessed as similar, above 95% [19]. Further, individual semen specimens from study participants were analyzed to determine viral persistence versus latency, as presented in Whitmer et al. [20].

A housekeeping gene sequence assessed specimen quality, including the presence of cellular beta 2 microglobulin (B2M) mRNA (Invitrogen), which served as an RNA extraction and sample authenticity control.

### Outcome variable

The outcome event of analysis was a confirmed negative result, defined as 2 consecutive EBOV RNA negative semen specimens. A cutoff Ct value of 40 was used to define positive versus negative detection in both assays.

Algorithms of interpretation, including for specimen quality, were developed to categorize results into positive, negative, or missing. PCR results were considered (1) missing in cases of no detection of housekeeping gene and target gene sequences, in which case they were handled as missing values in the analyses; (2) EBOV RNA negative when the housekeeping gene was detected but none of the 2 target gene sequences; (3) EBOV RNA positive when both target gene sequences were detected; and (4) EBOV RNA indeterminate when only one target was

detected and the other one not. In the dichotomized outcome analyses, indeterminate results were included in the positive category.

## Representativity of sample—link to the Sierra Leone national Ebola database

To assess the representativeness of our convenience sample and the possibility of selection bias, the study population was compared to the Sierra Leone national EVD survivor database. The database has been compiled through active searches in Port Loko and Kambia, and mass registrations at survivor conferences in all other districts, and includes 919 adult male survivors. Variables for patient demographics, including age at the time of EVD, occupation, and marital status, were extracted for the comparison.

## Variables

**Sociodemographic characteristics, symptoms during acute infection, and EVD sequelae.** Independent variables included age, education, marital status, household size, relationship to other household members, whether any other household members had EVD, sexual behavior, and adherence to preventive sexual advice after ETU discharge.

Participants reported on the presence of the following during acute EVD: vomiting, diarrhea, being unable to get up to go to the toilet, and being unable to drink by themselves. EVD sequelae (involving eyes, joints, neurological system, abdomen, skin, mental health, and sexual function) were reported at baseline and at every follow-up visit, and any sequelae reported in the 24 months after ETU discharge were used.

**Acute EVD Ct values.** Associations between persistence duration of EBOV RNA in semen and viremia during acute EVD were analyzed by abstracting acute blood qRT-PCR results from the WHO global laboratory database in Sierra Leone, including multiple blood Ct value tests over time for the same patient [21]. A variable was constructed using the matched individual's lowest Ct value available as a proxy for maximum viral load: 121 participants could be matched and were included in this subset. The Ct value variable was dichotomized, based on the group mean acute Ct value of 27.9 with a threshold set at Ct value $\geq 27$. Higher values indicate lower quantities of circulating EBOV RNA during acute disease, i.e., milder disease, and lower Ct values indicate more EBOV RNA present, i.e., more severe disease [22].

**Survival analysis.** The Kaplan–Meier interval censoring method was used to calculate the rate of EBOV persistence in semen at different time points starting from ETU discharge. Participants negative at baseline were interval-censored (IC) between the ETU discharge date and date of first baseline testing. Participants who were positive at recruitment were followed every 2 weeks, and considered IC between the dates of their last positive and first confirmed negative result. Survivors who did not reach the endpoint were right-censored, as in leaving the cohort before the event of interest occurred.

A parametric Weibull model was fitted using the IC data and superimposed on the Kaplan–Meier IC curve for the overall sample. The Weibull persistence rates were based on expected times from ETU discharge to EBOV negativity. Interval censoring within the univariate and multivariable semi-parametric IC proportional hazard models was estimated by the SAS ICPHREG procedure [23]. The multivariate model building strategy that gave rise to the final multivariate model, including the considerations for possible interaction effects, is discussed in S1 Statistical Analysis Plan based on methodologies in [24]. Interaction terms were only included if they were highly significant, with $p$-value $< 0.01$. The 95% confidence intervals were based on the complementary log–log transformation. Hazard ratio (HR) $> 1$, HR $< 1$, and HR $= 1$ denoted a higher probability, a lower probability, and no difference in the

probability of being confirmed negative for EBOV RNA (i.e., a relatively lower, higher, and similar risk of EBOV persistence in semen), respectively. A 2-sided $p$-value < 0.05 was considered statistically significant.

The prospective study protocol and the prospective statistical analysis plan are provided as S1 Protocol and S1 Statistical Analysis Plan, respectively. The study benefitted from an independent data monitoring committee advising on the conduct, analyses, and dissemination of the study.

### Ethics

Ethical permission was granted from the Sierra Leone Ethics and Scientific Review Committee and the WHO Ethical Review Committee (No. RPC736).

This study is reported as per the Strengthening the Reporting of Observational Studies in Epidemiology (STROBE) guideline (S1 STROBE Checklist).

## Results

### Study participants and retention

Out of the 220 men enrolled, 203 were included in the analyses while 17 men were excluded because of: no semen sample collected (n = 10), only one sample collected (n = 3), found not infected with Ebola virus (n = 4). Fifty-seven participants presented with at least 2 EBOV RNA positive semen specimens; 37 had, at least once prior to the endpoint, a negative test consecutively followed by a positive test. Of the 7 who were right-censored, 4 had at least once a single negative result that was consecutively followed by a positive result.

### Baseline participants' characteristics and uptake of safe sex advice

The median time between ETU discharge and study admission was 258 days (range: 40–610 days). Baseline characteristics are summarized in Table 1 and have been presented in detail elsewhere [13].

Sixty-five of the 220 men (30%) reported having had sexual contact during the 3 months following ETU discharge, and more than half of those (54%) reported condom use only sometimes or not at all (Table 1).

### EBOV persistence detected in semen over time

At 180 days (6 months) after ETU discharge, the EBOV RNA semen positive rate was 75.4% (95% CI 66.9%–82.0%). The median persistence duration was 204 days, with 50% of men having cleared their semen of EBOV RNA by this time. After 270 days, persistence was 26.8% (95% CI 20.0%–34.2%) and at 360 days, 6.0% (95% CI 3.1%–10.2%) (Table 2; Fig 1). The maximum duration of persistence in semen observed in this cohort was 696 days following ETU discharge.

### Severe acute disease and older age associated with EBOV RNA persistence in semen

Severe disease, measured by lower acute EVD blood Ct value, was significantly correlated with longer EBOV RNA persistence. The probability of persistence at 1 year after ETU discharge was 10.1% (95% CI 4.6%–19.8%) and approached 0% only after 720 days. Among participants with mild acute disease, the decay curve fell steeply to approach 0% already at 1 year (Fig 2).

**Table 1. Baseline characteristics of EVD survivors enrolled in the study.**

| Characteristics | Pooled (*N* = 220) |
|---|---|
| **Days from ETU discharge to study admission** | |
| Mean (SD) | 299 (141) |
| Median (IQR) | 258 (187, 422) |
| Minimum, maximum | 40, 610 |
| **Age (years)** | |
| Mean (SD) | 31.5 (9.5) |
| Median (IR) | 29.0 (25.0, 36.0) |
| Minimum, maximum | 18.0, 66.0 |
| ≤25 | 67/219 (30.6) |
| 26–35 | 91/219 (41.6) |
| >35 | 61/219 (27.8) |
| *Missing* | *1* |
| **Highest level of education** | |
| No education | 43/220 (19.6) |
| Primary/1–8 years of education | 70/220 (31.8) |
| Secondary or higher/8+ years of education | 107/220 (48.6) |
| **Marital status** | |
| Single/divorced/widowed/separated | 98/220 (44.6) |
| Married/long-term relationship | 122/220 (55.4) |
| **Cohabitants in household** | |
| Spouse or partner | 76/219 (34.7) |
| Parents | 52/219 (23.7) |
| Extended family | 131/219 (59.8) |
| **Household size (number of persons including self)** | |
| ≤4 | 56/216 (25.9) |
| 5–8 | 73/216 (33.8) |
| 9–12 | 43/216 (19.9) |
| >12 | 44/216 (20.4) |
| *Missing* | *4* |
| **Number of household members with EVD (excluding self)** | |
| None | 58/220 (26.4) |
| 1–2 | 61/220 (27.7) |
| 3–4 | 42/220 (19.1) |
| 5+ | 59/220 (26.8) |
| *Missing* | |
| **Symptoms during EVD** | |
| Vomited | 174/219 (79.5) |
| Too sick to relieve self on toilet | 109/219 (49.8) |
| Diarrhea | 174/219 (79.5) |
| Too sick to drink for more than a day | 102/219 (46.6) |
| **Overall health and well-being now as compared to before EVD** | |
| Back to normal or same as before | 81/218 (37.2) |
| Worse than before | 72/218 (33.0) |
| Better than before | 65/218 (29.8) |
| **New health problems since recovering from EVD** | 134/220 (60.9) |
| **Symptoms experienced any time from discharge to 24 months** | |
| Eye/vision problems | 58/220 (26.4) |

(*Continued*)

**Table 1.** (Continued)

| Characteristics | Pooled (*N* = 220) |
|---|---|
| Joint pain | 116/220 (52.7) |
| Psychological problems (including anxiety and depression) | 9/220 (4.1) |
| Sexual problems/sexual desire not same as before | 41/220 (18.6) |
| **Tested for HIV** | 143/216 (66.2) |
| *Missing* | *4* |
| **Sexual behavior during the first 3 months after recovering from EVD** | |
| Had sex | 65/220 (29.5) |
| Sexual desire following EVD recovery was the same as before | 167/220 (75.9) |
| Condom use | |
| Never | 20/219 (9.1) |
| Sometimes | 15/219 (6.8) |
| All the time | 28/219 (12.8) |
| Not sexually active | 156/219 (71.2) |
| *Missing* | *1* |
| **Since EVD recovery, difficulty getting or maintaining erection or ejaculating** | 56/220 (25.5) |
| **Blood Ct value during ETU admission** | |
| <27 | 73/127 (57.5) |
| ≥27 | 54/127 (42.5) |

Data are *n/N* (%), unless otherwise indicated.

Ct, cycle threshold; ETU, Ebola treatment unit; EVD, Ebola virus disease; IQR, interquartile range; SD, standard deviation.

Further, age was significantly associated with EBOV RNA persistence in the multivariate regression model. Showing a dose–response pattern, the youngest men (≤25 years) were 3.4 times (95% CI 2.0, 5.8; $p < 0.001$) more likely to be EBOV RNA negative in semen, and men aged 26–35 were 1.7 times (95% CI 1.1, 2.6; $p = 0.013$) more likely, as compared to men aged >35 years, at any given point in time (Table 3).

The age-stratified survival models revealed an effect modification of age with acute blood Ct value and persistence, where in the group of survivors who had severe disease at admission (blood Ct value < 27), age older than 25 years was associated with longer persistence

**Table 2. Kaplan–Meier non-parametric interval-censored probability estimates of Ebola virus RNA persistence in semen.**

| Time from ETU discharge (days) | EBOV persistence probability (95% CI) |
|---|---|
| 0 | — |
| 60 | 96.0 (90.2, 98.4) |
| 90 | 92.1 (85.2, 95.8) |
| 180 | 75.4 (66.9, 82.0) |
| 270 | 26.8 (20.0, 34.2) |
| 360 | 6.0 (3.1,10.2) |
| 450 | 5.1 (2.5, 9.0) |
| 540 | 2.0 (0.6, 4.9) |

EBOV, Ebola virus; ETU, Ebola treatment unit.

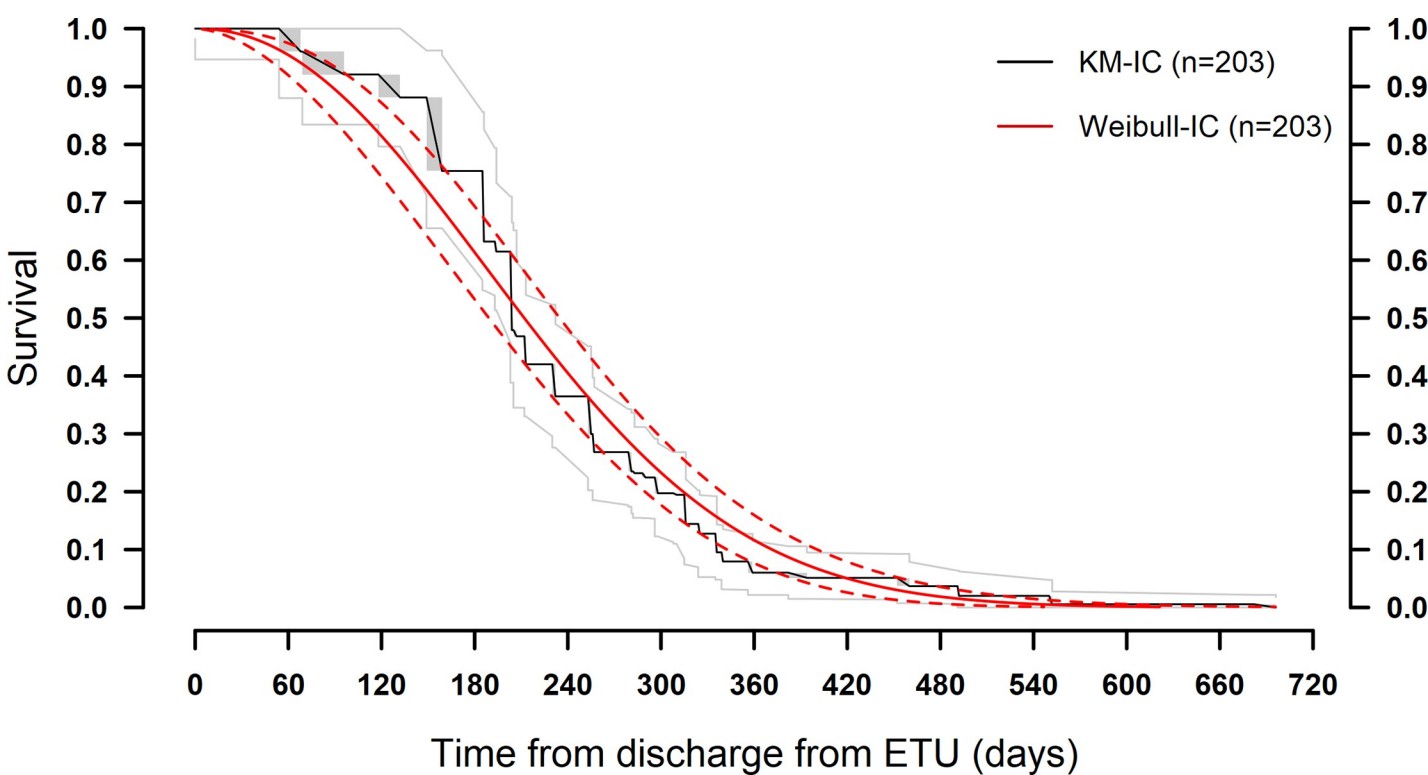

**Fig 1. Survival function (Kaplan–Meier analysis) of Ebola virus RNA persistence in semen over time since ETU discharge, with Weibull fit.** ETU, Ebola treatment unit; IC, interval-censored; KM, Kaplan–Meier.

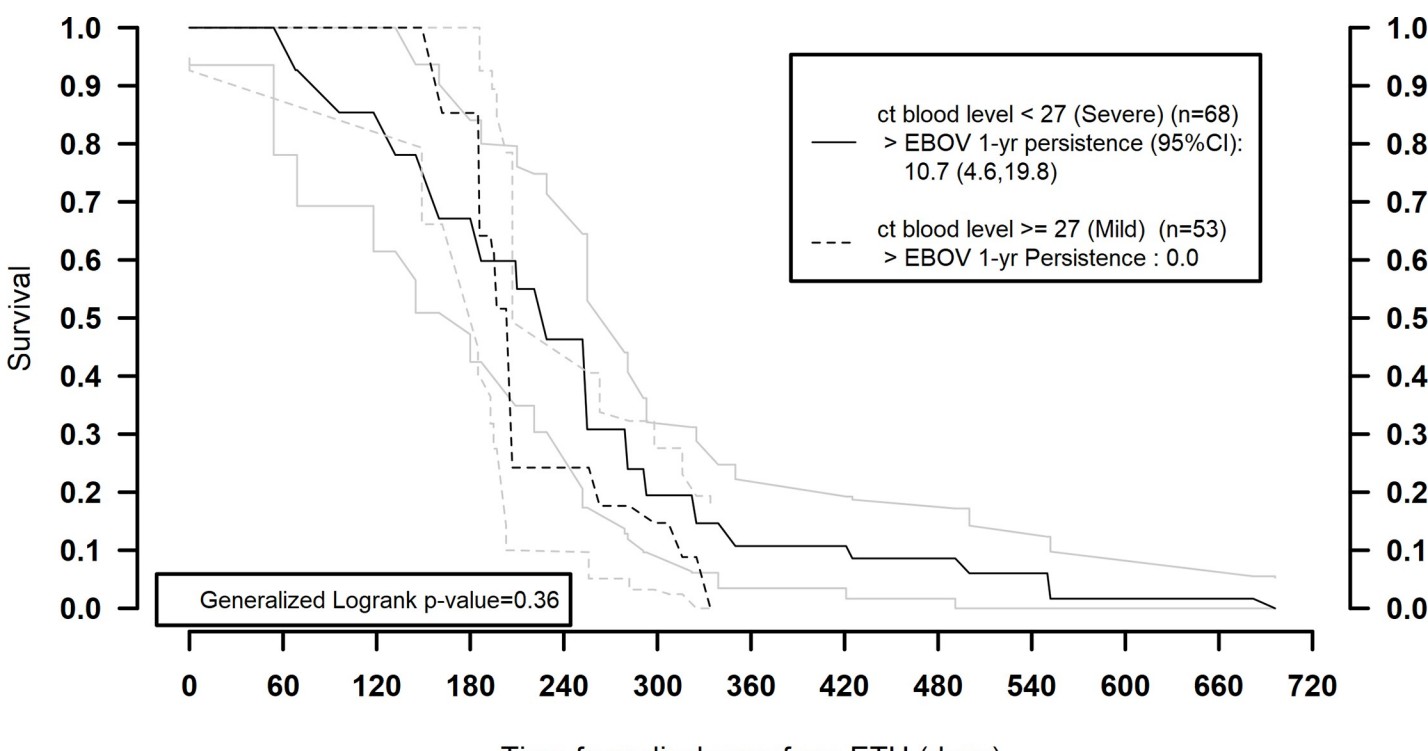

**Fig 2. Survival function of EBOV RNA persistence stratified by acute/ETU blood cycle threshold values, within the sub-population with matched values available.** ct, cycle threshold; EBOV, Ebola virus; ETU, Ebola treatment unit.

**Table 3. Multivariate model with living with extended family × education interaction term.**

| Factors | Adjusted HR (95% CI) | Chi-squared p-value |
|---|---|---|
| **Age (years)** | | |
| ≤25 | 3.44 (2.04, 5.81) | <0.001 |
| 26–35 | 1.70 (1.12, 2.59) | 0.013 |
| >35 | 1.00 | |
| **Five or more household members with EVD (excluding self)** | 0.56 (0.37, 0.84) | 0.005 |
| **Diarrhea symptoms during EVD** | 0.49 (0.30, 0.80) | 0.004 |
| **Education × living with extended family interaction term** | | 0.001 |
| Level 1: Not living with extended family—secondary or higher/8+ years of education (versus primary level or lower/<8 years of education) | 2.62 (1.50, 4.58) | |
| Level 2: Living with extended family—secondary or higher/8+ years of education (versus primary level or lower/<8 years of education) | 0.77 (0.49, 1.20) | |

HR > 1 implies higher probability of being confirmed EBOV RNA negative in semen (lower likelihood of EBOV persistence); HR < 1 implies a lower probability of being confirmed EBOV RNA negative in semen (increased likelihood of EBOV persistence).

CI, confidence interval; EBOV, Ebola virus; EVD, Ebola virus disease; HR, hazard ratio.

(compared to age ≤ 25 years). Twenty percent remained positive at 1 year in this group (generalized log-rank $p$ = 0.001) (Table 4; Fig 3). For those with milder acute disease (blood Ct value ≥ 27), persistence was not significantly different between the 2 age groups (generalized

**Table 4. Multivariate model in a subset of 121 participants with available ETU blood Ct value.**

| Factors | Adjusted HR (95% CI) | Chi-squared p-value |
|---|---|---|
| **Age, (years)** | | |
| ≤25 | 3.17 (1.60, 6.29) | 0.001 |
| 26–35 | 1.85 (1.04, 3.28) | 0.036 |
| >35 | 1.00 | |
| **ETU blood Ct value < 27** | 0.61 (0.38, 0.98) | 0.043 |
| **Five or more household members with EVD (excluding self)** | 0.60 (0.35, 1.03) | 0.063 |
| **Diarrhea symptoms during EVD** | 0.48 (0.26, 0.88) | 0.018 |
| **Education × living with extended family interaction term** | | 0.006 |
| Level 1: Not living with extended family—secondary or higher/8+ years of education (versus primary level or lower/<8 years of education) | 2.63 (1.30, 5.34) | |
| Level 2: Living with extended family—secondary or higher/8+ years of education (versus primary level or lower/<8 years of education) | 0.71 (0.39, 1.27) | |

HR > 1 implies higher probability of being confirmed EBOV RNA negative in semen (lower likelihood of EBOV persistence); HR < 1 implies a lower probability of being confirmed EBOV RNA negative in semen (increased likelihood of EBOV persistence).

Ct, cycle threshold; EBOV, Ebola virus; ETU, Ebola treatment unit; EVD, Ebola virus disease; HR, hazard ratio.

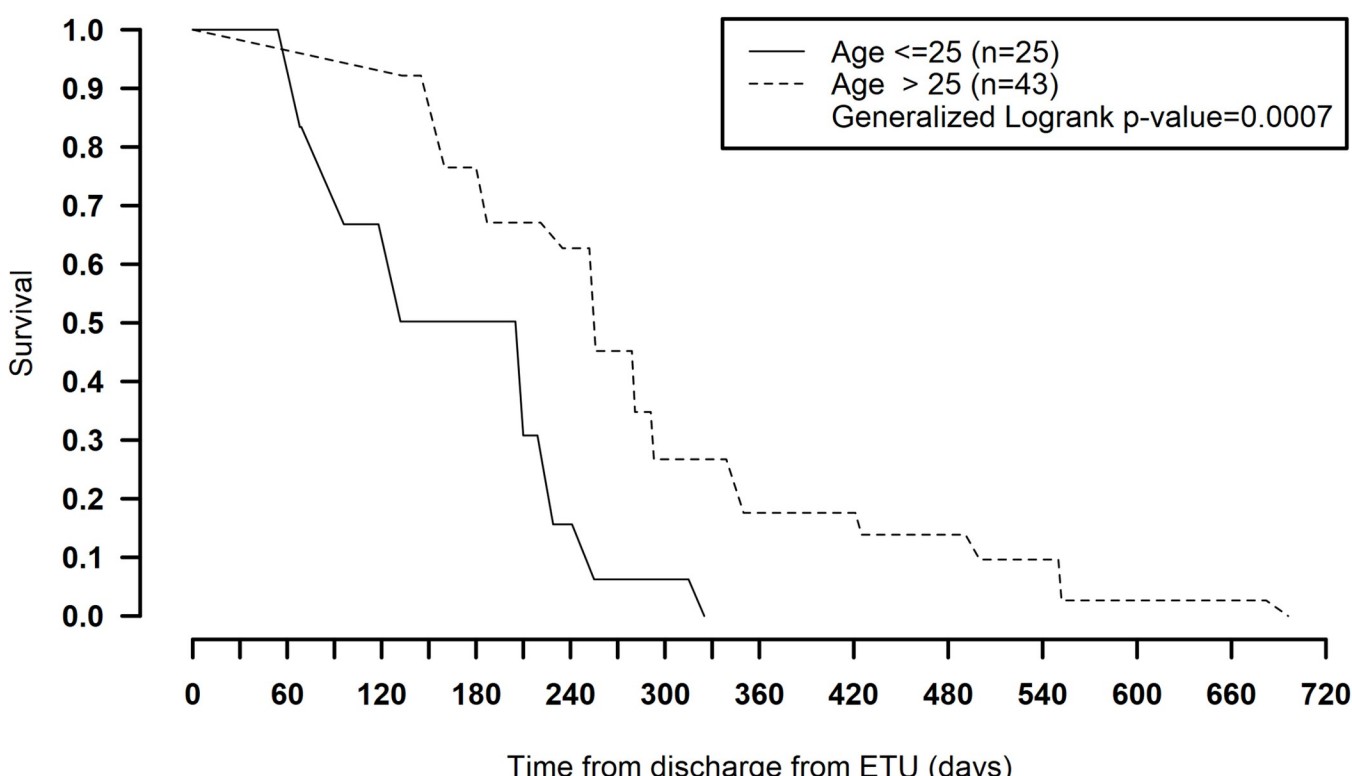

**Fig 3. Survival function of Ebola virus RNA persistence over time (days) stratified by age, within the sub-population who had severe acute disease (ETU cycle threshold value < 27).** ETU, Ebola treatment unit.

log-rank $p = 0.79$) (Fig 4). Also, reporting diarrhea during acute disease was significantly associated with longer persistence in semen (Table 3).

## EVD co-morbidity and EVD sequelae

None of the men self-reported co-morbidity for tuberculosis, diabetes, malaria, or sexually transmitted infections at the time of their acute infection with EVD. Among the 143 men tested for HIV at enrollment, 1 tested HIV positive; his semen specimen tested negative for EBOV RNA.

More than half of the participating survivors (60%) reported new health problems following their acute EVD. Symptoms involving the joints, followed by symptoms of the eye/vision and lack of sexual desire or impotency, were most commonly reported (Table 1).

## Representativity of study sample

A comparison of study participant demographics with those of registered male EVD survivors 18 years or above in Sierra Leone ($n = 919$) indicated minimal differences across comparable demographic indicators. In particular, there was a slightly higher proportion of men 26–35 years old recruited to the study (40%) compared to men of the same age group in the national EVD survivor database (34%). Additionally, a greater proportion of study participants were married or in a long-term relationship (56% versus 44%), and a smaller proportion reported not working (2% versus 22%), compared to those in the national registry. Due to limitations in the use of the national EVD survivor database, further statistical comparisons could not be conducted.

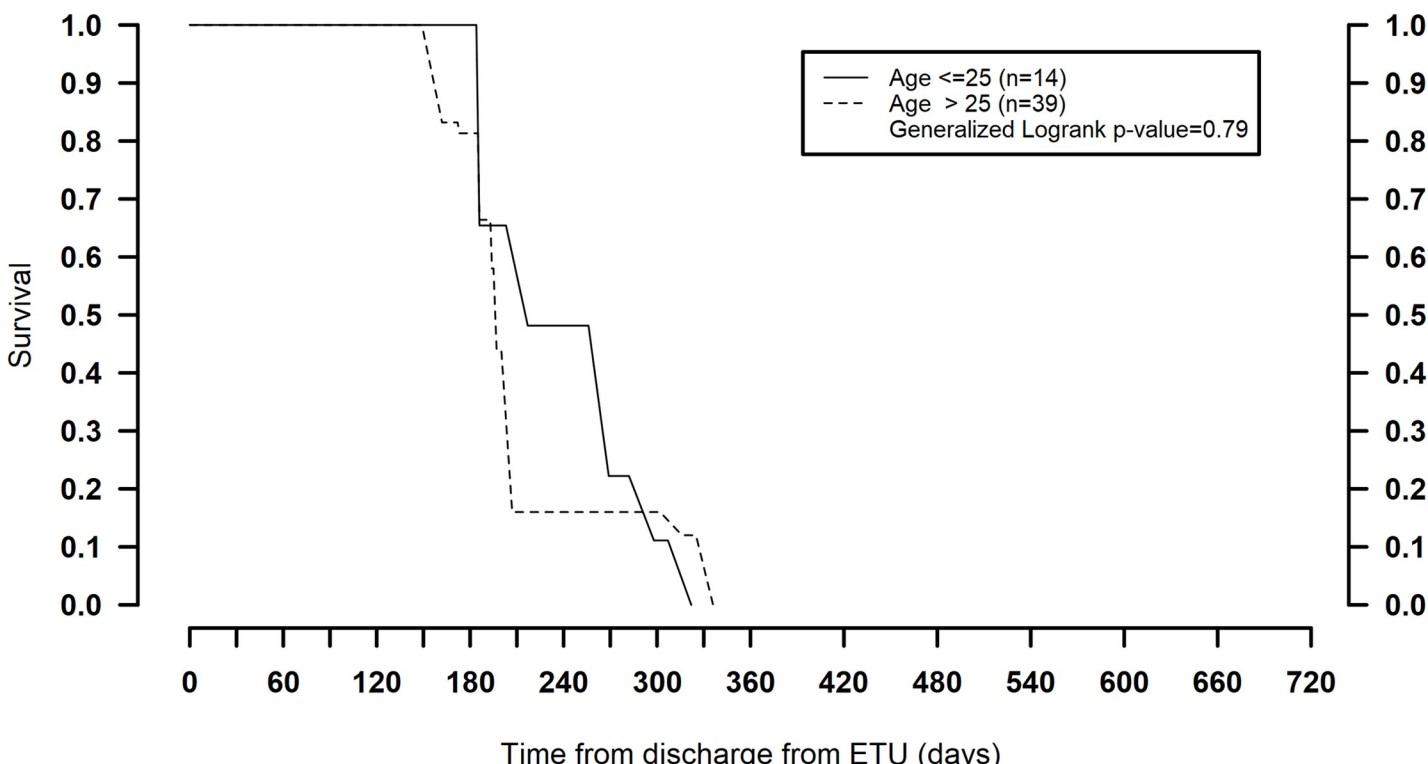

**Fig 4. Survival function of Ebola virus RNA persistence over time (days) stratified by age, within the sub-population who had milder acute disease (ETU cycle threshold value $\geq$ 27).** ETU, Ebola treatment unit.

## Discussion

Our findings show that in a cohort of 220 male EVD survivors from Sierra Leone, 75% of men were still semen positive for EBOV RNA 6 months after being discharged following acute EVD, and 50% at 204 days. At 1 year, less than 10% of study participants overall, but 20% of men aged >25 years who had had severe acute disease, had detectable EBOV RNA in semen. Longer persistence was significantly associated with increased age and severe acute EVD. The EBOV RNA positive rate in semen was considerably higher over time than what has previously been described. Earlier estimates of persistence rates among survivors based on longitudinal observations and modeling show lower estimates of EBOV RNA persistence rates. One study found persistence of 50% at 3.8 months [11] as compared to 50% remaining positive at close to 7 months in our cohort. Modeled estimates from the Postebogui cohort in Guinea, involving 27 men who tested positive in semen at least once, suggested a median duration of 45 days. In a later estimate by Keita et al. [11], the same population was estimated to have a 13%–60% probability of persistence at 6 months, but the probability was highly dependent on variations between different RT-PCR methods involved [11,25,26]. These studies provide important insights into findings from individual cases, followed up with different and sometimes long intervals. Our study involves a larger sample and does not rely on modeling outside of the survival analysis to estimate the population rates. Our finding of a 75% probability of persistence at 6 months hence represents an empirical longitudinal cohort analysis including regular, frequent follow-up and a large sample, which can explain the significant differences to earlier estimates.

The maximum duration of EBOV RNA positivity observed, 696 days, confirms published findings on EBOV RNA presence in semen from analyses of cross-sectional sample composition or case reports of maximum duration of EBOV RNA detection, including the baseline analyses of this study [9,12,13,27–29]. The higher than expected persistence over time we find in this large cohort of survivors, with 75% of men having EBOV RNA present at 6 months, sheds light on hypotheses raised by these studies, by here adding analyses of persistence rates over time from a representative male survivor sample.

We identified low adherence to initial safe sex counseling among 30% of the participants. In combination with our findings of higher than expected rates of EBOV RNA persistence in semen over time, the risk of residual sexual transmission is present. These results represent critical information to inform Ebola epidemic preparedness and response. The impact of a single case of sexually transmitted EBOV infection can be devastating, to both an individual and to public health, as evidenced by the case of sexual transmission 470 days after ETU discharge reported from Guinea [7]. Inconsistent condom use and limited compliance with the initial ETU recommendation of abstinence are identified challenges that merit immediate action to understand barriers to and facilitators of sustained risk reduction.

Our results further highlight the urgent need to ensure efforts to organize a national response providing semen testing, safe sex counseling, and free access to condoms, as a priority from the start of an outbreak. The here included results will inform forthcoming WHO updated guidelines on semen testing and will contribute to raising awareness of semen program testing needs in the context of the ongoing outbreak in DRC. Most semen testing needs would occur during the first year following ETU discharge, with half of the survivors expected to remain qRT-PCR positive up until close to 7 months. Based on our results, early risk identification and targeted counseling on long-term persistence risks among survivors, especially among those >25 years and with higher quantities of EBOV RNA in blood during acute disease, could be part of efforts to motivate safe sex and retain survivors in the testing program. These efforts will also need to be sustained, together with vigilance and responsiveness in the aftermath of the acute epidemic phase. The risk of new clusters of EVD igniting through sexual transmission demands a thorough and complex epidemic response over time, where a sustainable semen testing program is a crucial component.

Participants in our study who were older and those who had severe acute disease, defined as acute blood Ct value < 27, and those with diarrhea during acute EVD had a significantly higher probability of having persistent EBOV RNA in semen over time. Twenty percent of men with severe disease and >25 years had detectable EBOV RNA in semen at 1 year. Overall, younger age seemed to reduce the time to semen EBOV RNA clearance by dose–response pattern. This confirms hypotheses raised from other studies of survivors relating to men >40 years; we show how the risk increase of longer persistence starts at an earlier age than earlier anticipated [9,12]. Several mechanisms, such as immune response to acute viremia, natural aging of the immune system, or inherent co-morbidity, may explain these associations.

The sociodemographic links to persistence merit further in-depth bio-behavioral research.

We found an unexpectedly low frequency of co-morbidity associated with acute EVD, which may be related to higher EVD case fatality rates in patients with additional diagnoses. Similar to other reports on Ebola survivors, the men in our study reported a high frequency (>60%) of post-EVD sequelae including symptoms of sexual fatigue [27]. These results highlight that survivors need continued care efforts, also addressing sexual health.

In this study we opted to use 2 consecutive negative qRT-PCR results, collected 2 weeks apart, as the definition of negativity, but keeping the "time of event" to the first test and applying interval censoring as needed, to accurately reflect time to event. For semen positive participants found positive at baseline, we assumed continuous positivity from ETU discharge until the first test at baseline.

ETU discharge date per certificate was used as the starting point for semen positivity. This date is a proxy for when viremia waned and will likely have underestimated persistence time. Acute Ct values in our study were not available for all participants, were taken at different points in time during acute disease, and were analyzed by different laboratories, which may have diluted the associations observed. The lowest available acute Ct value was consequently retrieved, to mitigate misclassification.

It should also be noted in all interpretations that the qRT-PCR detection of EBOV RNA does not distinguish between viable virus and RNA fragments. In an analysis by Whitmer et al. [20], including positive semen specimens from study participants in this cohort as well as from survivors in the US, it was shown that active EBOV replication occurred, and they concluded that "EBOV persistence within EVD survivors may act as a viral reservoir," supporting the relevance of a positive qRT-PCR finding. [20].

Our study applied purposive sampling to recruit participants. The comparison of study participants with registered male survivors 18 years or above in Sierra Leone indicated differences across comparable demographic indicators of marriage and unemployment. These differences could reflect selection bias, influencing the generalizability of the results, but these traits were adjusted for as covariates in the multivariate analysis of associations with the outcome.

Recall bias may influence validity, specifically in the questions about the acute disease episode and sexual behavior. This and social desirability in answering may have lowered estimates of risky sexual behavior.

## Conclusions

Our findings showed probabilities of semen persistence of EBOV RNA to be 75% at 6 months after ETU discharge, with persistence of 50% at 204 days and less than 10% at 1 year after ETU discharge. Persistence however remained at more than 20% at 1 year among participants >25 years with higher quantities of EBOV RNA in blood during acute disease. Uptake of safer sex recommendations 3 months after ETU discharge was low among a third of survivors. The study population was largely representative of the male EVD survivor population in Sierra Leon, apart from noted differences in marriage status and employment. These variables were adjusted for in the multivariate analysis, and we conclude that our results can be generalized to the wider male survivor population in Sierra Leone, and can also inform management and response to survivors' needs following EVD in other contexts.

Our results highlight the immediate needs of planning for increased vigilance and efforts to support male survivors with safe sex counseling, including free provision of condoms at discharge, together with implementation of a semen testing program, as part of epidemic preparedness and primary and sustained epidemic response. Evidence exists showing sexual transmission chains after recovery; however, more research is needed to better understand the contribution of sexual transmission at different points in an epidemic. Emerging EVD cases in the aftermath of an epidemic, as was the case in DRC, also merit in-depth analysis to understand the role of EBOV RNA semen persistence in survivors [30].

## Supporting information

**S1 STROBE Checklist.**
(DOCX)

**S1 Protocol. Study protocol for the EBOV persistence study version 1.7.** 25 January 2016.
(DOC)

**S1 Statistical Analysis Plan.**
(PDF)

## Acknowledgments

This work has been developed on behalf of the Sierra Leone Ebola Virus Persistence Study Group (for listing please revert to https://journals.plos.org/plosntds/article?id=10.1371/journal.pntd.0005723#ack). The investigators are grateful for advice provided by Ian Crozier, Johan Giesecke, Michael Hughes, Nicolas Meda, Janus Paweska, Donna Spiegelman, and Edith Tarimo in their role as expert advisors of the study's independent data monitoring committee.

## Author Contributions

**Conceptualization:** A. E. Thorson, G. F. Deen, K. T. Bernstein, K. N. Durski, P. Formenty, B. Knust, N. Broutet.

**Data curation:** N. Habib, P. Gaillard, E. Ervin, J. E. Marrinan.

**Formal analysis:** A. E. Thorson, N. Habib, P. Gaillard.

**Funding acquisition:** A. E. Thorson, B. Knust, N. Broutet.

**Investigation:** A. E. Thorson, G. F. Deen, W. J. Liu, F. Yamba, F. R. Sesay, T. A. Massaquoi, S. L. R. McDonald, Y. Zhang, K. N. Durski, S. Singaravelu, E. Ervin, J. E. Marrinan, P. Formenty, U. Ströher, W. Xu, B. Knust, N. Broutet.

**Methodology:** A. E. Thorson, K. T. Bernstein, N. Habib, N. Broutet.

**Project administration:** A. E. Thorson, G. F. Deen, K. T. Bernstein, F. Yamba, F. R. Sesay, P. Gaillard, T. A. Massaquoi, S. L. R. McDonald, K. N. Durski, S. Singaravelu, E. Ervin, H. Liu, A. Coursier, J. E. Marrinan, A. Ariyarajah, M. Carino, M. Lamunu, G. Wu, B. Knust, N. Broutet.

**Resources:** A. E. Thorson, G. F. Deen, F. R. Sesay, P. Gaillard, T. A. Massaquoi, S. L. R. McDonald, Y. Zhang, K. N. Durski, H. Liu, A. Coursier, A. Ariyarajah, F. Sahr, B. Knust, N. Broutet.

**Software:** N. Habib, S. Singaravelu.

**Supervision:** A. E. Thorson, G. F. Deen, K. T. Bernstein, W. J. Liu, F. Yamba, F. R. Sesay, P. Gaillard, T. A. Massaquoi, S. L. R. McDonald, Y. Zhang, K. N. Durski, S. Singaravelu, E. Ervin, A. Coursier, J. E. Marrinan, A. Ariyarajah, M. Carino, P. Formenty, U. Ströher, M. Lamunu, G. Wu, F. Sahr, W. Xu, B. Knust, N. Broutet.

**Validation:** P. Gaillard, S. L. R. McDonald, K. N. Durski, S. Singaravelu, E. Ervin, A. Coursier, J. E. Marrinan, A. Ariyarajah.

**Writing – original draft:** A. E. Thorson.

**Writing – review & editing:** A. E. Thorson, G. F. Deen, K. T. Bernstein, W. J. Liu, F. Yamba, N. Habib, F. R. Sesay, P. Gaillard, T. A. Massaquoi, S. L. R. McDonald, Y. Zhang, K. N. Durski, S. Singaravelu, E. Ervin, H. Liu, A. Coursier, J. E. Marrinan, A. Ariyarajah, M. Carino, P. Formenty, U. Ströher, M. Lamunu, G. Wu, F. Sahr, W. Xu, B. Knust, N. Broutet.

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
