## [Editor Report · Decision Letter 0]

10 Feb 2020

Dear Dr Thorson, 

Thank you for submitting your manuscript entitled "Persistence of Ebola virus in semen - A cohort study among survivors in Sierra Leone" for consideration by PLOS Medicine.

Your manuscript has now been evaluated by the PLOS Medicine editorial staff and I am writing to let you know that we would like to send your submission out for external peer review.

Kind regards,

Helen Howard, for Clare Stone PhD 

Acting Editor-in-Chief

PLOS Medicine 

plosmedicine.org

---

## [Decision Letter · Decision Letter 1]

27 Mar 2020

Dear Dr. Thorson,

Thank you very much for submitting your manuscript "Persistence of Ebola virus in semen - Risk factors, magnitude and duration, results from a longitudinal cohort study of survivors in Sierra Leone" (PMEDICINE-D-20-00300R1) for consideration at PLOS Medicine. 

[LINK]

In light of these reviews, I am afraid that we will not be able to accept the manuscript for publication in the journal in its current form, but we would like to consider a revised version that addresses the reviewers' and editors' comments. Obviously we cannot make any decision about publication until we have seen the revised manuscript and your response, and we plan to seek re-review by one or more of the reviewers. 

We expect to receive your revised manuscript by Apr 17 2020 11:59PM. Please email us (plosmedicine@plos.org) if you have any questions or concerns.

We look forward to receiving your revised manuscript. 

Sincerely,

Adya Misra, PhD

Senior Editor 

PLOS Medicine

plosmedicine.org

Title-Please revise your title according to PLOS Medicine's style. Your title must be nondeclarative and not a question. It should begin with main concept if possible. "Effect of" should be used only if causality can be inferred, i.e., for an RCT. Please place the study design ("A randomized controlled trial," "A retrospective study," "A modelling study," etc.) in the subtitle (ie, after a colon).

Abstract

Please structure your abstract using the PLOS Medicine headings (Background, Methods and Findings, Conclusions).

Background- Provide the context of why the study is important. The final sentence should clearly state the study question.

Please ensure that all numbers presented in the abstract are present and identical to numbers presented in the main manuscript text. * Please include the study design, population and setting, number of participants, years during which the study took place, length of follow up, and main outcome measures. * Please quantify the main results (with 95% CIs and p values). * Please include the important dependent variables that are adjusted for in the analyses. * Please include the actual amounts and/or absolute risk(s) of relevant outcomes (including NNT or NNH where appropriate), not just relative risks or correlation coefficients. (example for absolute risks: PMID: 28399126). * Please include a summary of adverse events if these were assessed in the study. * In the last sentence of the Abstract Methods and Findings section, please describe the main limitation(s) of the study's methodology/study design.

Abstract Conclusions:

* Please address the study implications without overreaching what can be concluded from the data; the phrase "In this study, we observed ..." may be useful.

* Please interpret the study based on the results presented in the abstract, emphasizing what is new without overstating your conclusions.

* Please avoid vague statements such as "these results have major implications for policy/clinical care". Mention only specific implications substantiated by the results.

* Please avoid assertions of primacy ("We report for the first time....")

The Data Availability Statement (DAS) requires revision. For each data source used in your study: 

Please use Vancouver style for references and place all references within square brackets

Author summary

Introduction

Please address past research and explain the need for and potential importance of your study. Indicate whether your study is novel and how you determined that. If there has been a systematic review of the evidence related to your study (or you have conducted one), please refer to and reference that review and indicate whether it supports the need for your study.

Please conclude the Introduction with a clear description of the study question or hypothesis.

Methods

Please provide details or citations of all interview guides used in the study. If these are not published, please provide a copy as SI files.

Please report the number of [patients, samples, etc] and dates of recruitment, and account for all methods used in your study.

Ethics approval- please provide details and name of the approving committee in the methods

Please ensure that the study is reported according to the STROBE guideline, and include the completed STROBE checklist as Supporting Information. Please add the following statement, or similar, to the Methods: "This study is reported as per the Strengthening the Reporting of Observational Studies in Epidemiology (STROBE) guideline (S1 Checklist)."

Did your study have a prospective protocol or analysis plan? Please state this (either way) early in the Methods section.

c) In either case, changes in the analysis-- including those made in response to peer review comments-- should be identified as such in the Methods section of the paper, with rationale.Please provide references to databases used in the study

Please provide further details about the housekeeping gene used in this study, along with the sequence as supplementary information 

Page 6 footnote-please move this to main text

Is it necessary to mention FAM and VIC reporter dyes on page 7 without any background? Please revise so that the methods can be read and understood by a general audience

Please provide details about when this study was carried out, when were samples collected and details of follow-up. 

PLOS does not permit "data not shown.” Please remove this claim, or do one of the following: a) If you are the owner of the data relevant to this claim, please provide the data in accordance with the PLOS data policy, and update your Data Availability Statement as needed. b) If the data not shown refer to a study from another group that has not been published, please cite personal communication in your manuscript text (it should not be included in the reference section). Please provide the name of the individual, the affiliation, and date of communication. The individual must provide PLOS Medicine written permission to be named for this purpose. c) For any other circumstance, please contact me ASAP.

Discussion

Please present and organize the Discussion as follows: a short, clear summary of the article's findings; what the study adds to existing research and where and why the results may differ from previous research; strengths and limitations of the study; implications and next steps for research, clinical practice, and/or public policy; one-paragraph conclusion.

Comments from the reviewers:

Reviewer #1: In the proposed paper, the authors present the follow-up of semen of Ebola survivors. They find that Ebola virus was present in many individuals, even a year after follow-up. This presents an interesting follow-up to the original paper (Deen et al., NEJM) as it includes some longitudinal data not included in the original paper. Nevertheless I was confused by the statistical approach taken.

My major concern/confusion is with the main regression analysis. The authors conduct a multivariable Weibull regression, where they include an interaction term between education and living with extended family. They also dedicate an entire table (Table 3.2) to stratified results for living and not living with extended family. The rationale for this approach was not set out in the introduction or methods and only briefly mentioned in the results. I also struggle to come up with a mechanistic rationale as to why this would be of particular interest. In the discussion there is a mention of reinfection risk over time, but then again, I struggle with why living with extended family is a good marker of this. I think this focus is an unfortunate distraction for the paper and I would consider removing it.

The 95% confidence intervals for the Kaplan-Meier curve (Figure 1) look very narrow, especially as there are only 220 individuals. The authors should clarify how these were calculated. The authors should also include 95% confidence intervals for the Weibull fit in Figure 1.

There are no uncertainty intervals presented in Figure 2 and Figure 3 - these need to be included.

The authors refer to 'data not shown' to say their sample is representative of the wider population. There is sufficient space in the article to set out the comparisons between their cohort and the wider population in a table - and show how the two populations are different and the same.

The authors use two sequential negative tests as a marker of being a true negative. It would be useful to set out how frequently a negative was followed up by a positive test to give readers an idea of how robust this definition is.

Lower CT score is used as a proxy for severe disease. The rationale for this proxy association should be referenced (e.g., Faye et al., PLoS Medicine). Viral load itself is of interest (irrespective if it is linked to symptoms), however, it will depend on when it was tested during acute illness - this weakness should be discussed.

On page 6, I do not understand the sentence 'A housekeeping gene sequence assessed specimens' quality through compatibility with human DNA' - the authors should clarify exactly what was done.

The authors use a simple dichotomization of greater/less that a CT of 27 for high/low disease severity - it would be useful to look at additional bins of CT (or use a spline) to see the full relationship between the two.

Reviewer #2: In this manuscript, Thorson et al analyzed the persistence of EBOV RNA in semen from a longitudinal cohort of survivors in Sierra Leone. Using two different EBOV RT-PCR, they monitored the kinetics of EBOV persistence in semen and established the survival function of EBOV RNA persistence over time according to time after discharge and to initial viremia and age. They demonstrated that the persistence correlated with age and other factors such as the occurrence of diarrhea during acute disease. The study is well designed and managed, the statistical analyses are robust, and the main strength of the study is the longitudinal follow-up in survivors with samples obtained very frequently. Some results obtained with this cohort have previously been published in NEJM by the authors and the current report provides the complete and final follow-up of the cohort.

The main weakness of this study is the lack of real novelty of the results, as a lot of reports describing the persistence of EBOV RNA in semen have already been published, with quite comparable conclusions. For example, the most recent publication (Keita, Open Forum Infect Dis), not cited by the authors, described a similar cohort with analysis of several body fluids including semen. Compared to the previous reports, no new discovery was provided in the current study. 

Major concerns.

It would have been interesting and important to perform viral isolation in addition to viral RNA detection to obtain information about the risk of EBOV sexual transmission according to the time after discharge. In addition, viral isolation would have allowed to perform genome sequencing of persisting viruses. These data would have constituted an important advance in the field.

The authors should have cited and discussed more exhaustively the existing literature about EBOV persistence such as Whitmer 2018 (Cell Reports 22, 1159), Keita 2019 (OFID 6,12), Barnes 2017 (CID 65:1400)…

Minor concerns.

In table 1, the information about the sexual desire appear twice, but with different values

Reviewer #3: The study by Thorson et al. addresses a public health problem for Ebola virus disease by providing a thorough, and very well written, analysis of samples and data rather difficult to achieve. I only suggest minor changes.

While the study provides academically sound results, their public health relevance could be improved by addressing some virological issues.

Results of the same study was already published earlier (Deen et al. 2017). Of 220 participants EBOV RNA was detected up to 1.5 years after ETU discharge. They stated, that "These data showed the long-term presence of Ebola virus RNA in semen and declining persistence with increasing time after ETU discharge." An added value of the published data could be achieved by evaluating and concluding directly the risk of sexual transmission of EVD. Barnes et al. (CID 2017:65) describe in their case report the isolation of viable EBOV in semen up to day 32 (after clinical recovery from EVD) and active replication in cells present in semen. Unfortunately, the submitted results does not relate to viable virus in semen but only viral RNA fragments. Although this might be difficult to achieve, it would strengthen any recommendation based on the presented data. By using an internal standard the amount of virus genome copy numbers could be used instead of PCR ct-values for a more informative conclusion and risk analysis. It has been shown by Barnes et al. that low ct-values of EBOV can be detected in semen within the first weeks/months after recovery from EVD. It would be helpful to correlate ct-values with amounts of viable virus (either directly from the clinical samples or by spiking experiments). Further, results are only defined as positive and negative by ct-values below or above 40. Depending on the PCR protocol a more informative outcome and conclusion could be achieved by presenting detailed ct-values or even better viral genome copy numbers. 

About the housekeeping gene target as internal control: was RNA being used after DNase treatment to avoid measuring DNA fragments? Please specify in methods section. Quality control for a RNA target should be undertaken by use of a RNA internal control.

Despite recommendations for safe sex, a third of survivors had a low uptake. Did you include testing of sex partners for monitoring of a possible transmission/infection? This could easily be tested by serology? This would further allow calculating or discussing the risk for sexual transmission and resulting infection with EBOV?

As the author's state in their study, a high proportion of male survivors regularly had unprotected sex after recovery from EVD, and obviously without causing another outbreak. How can this be explained?

Reviewer #4: Thorson and colleagues present a thorough analysis of the magnitude and duration of Ebola virus persistence in semen from survivors in Sierra Leone. The authors also include analysis of potential risk factors that influence persistence. 

Overall, I had few, if any, concerns with the manuscript. There were a few instances of redundant use of terms in the same or back-to-back sentences; however, these could be rectified with an additional thorough proofread.

The authors should be commended on their analysis. In particular, the authors provide evidence for more thorough behavioral guidance and analysis regarding sexual/reproductive health for EVD survivors. In addition, the authors have identified correlations between age and duration of persistence. As the authors point out, this could be related to a number of biological variables that should be further studied. 

Taken together, the authors have provided an informative manuscript that expands on their prior work while also providing guidance for public health practices.

[LINK]

---

## [Decision Letter · Decision Letter 2]

30 Jun 2020

Dear Dr. Thorson,

Thank you very much for re-submitting your manuscript "Persistence of Ebola virus in semen among survivors in Sierra Leone: A cohort study of frequency, duration and risk factors" (PMEDICINE-D-20-00300R2) for review by PLOS Medicine.

I have discussed the paper with my colleagues and the academic editor and it was also seen again by reviewers. I am pleased to say that provided the remaining editorial and production issues are dealt with we are planning to accept the paper for publication in the journal.

[LINK]

We look forward to receiving the revised manuscript by Jul 07 2020 11:59PM. 

Sincerely,

Adya Misra, PhD

Senior Editor 

PLOS Medicine

plosmedicine.org

Requests from Editors:

Please ensure that in the resubmitted version the main, revised doc is the only doc submitted in the file. This should be a clean doc with no tracking (I am finding it very difficult to know what is the final version and this is likely to add delays if it comes back again like this.) and also the shaded ‘confidential’ removed. When multiple copies are submitted in the revised file it is confusing as to which is the latest version. Any old or tracked versions can go into a supp file. 

Abstract “Leon” correct to Leone

Abstract -we need accurate dates (insert months) where participants were recruited from, how many there were and some summary demographic information including mean age. 

Abstract – “EVD Ct-values” on first mention, please define. Also ETU discharge

Abstract and throughout – p values need to be provided with 95% Cis for all quantifiable data

“Longer persistence was significantly associated” – please provide data and stats to support such statements. And again on page 16

Please check spellings (from Author Summary: Eevidence)

Refs should be presented in square not rounded brackets in the main text

PLOS Medicine requires that the de-identified data underlying the specific results in a published article be made available, without restrictions on access, in a public repository or as Supporting Information at the time of article publication, provided it is legal and ethical to do so. Please see the policy at http://journals.plos.org/plosmedicine/s/data-availability and FAQs at http://journals.plos.org/plosmedicine/s/data-availability#loc-faqs-for-data-policy

The email address you provide – who is that for? We cannot allow authors to be points of contact and we need to know what restrictions are in place for those who apply for access. It is not sufficient to say data is private 

Page 13 “right-censored” what does this mean?

Page 16 “None of the men reported co-morbidity in the forms of tuberculosis, diabetes, malaria or sexually transmitted infections (STI) in relation to their acute infection with EVD” please state clearly this is self reported and can be inaccurate as such and what do you mean by ‘in relation to’. Arent they independent ?

STROBE – please report using sections and paragraphs as pages change on revision / formatting.

Analysis plan – we asked if you had a prospective analysis plan – this is different to a statistical plan and thank you for providing that, but we still need an analysis plan or a discussion in the methods 

As a reminder: did you study have a prospective protocol or analysis plan? Please state this (either way) early in the Methods section. a) If a prospective analysis plan (from your funding proposal, IRB or other ethics committee submission, study protocol, or other planning document written before analyzing the data) was used in designing the study, please include the relevant prospectively written document with your revised manuscript as a Supporting Information file to be published alongside your study, and cite it in the Methods section. A legend for this file should be included at the end of your manuscript. b) If no such document exists, please make sure that the Methods section transparently describes when analyses were planned, and when/why any data-driven changes to analyses took place. c) In either case, changes in the analysis-- including those made in response to peer review comments-- should be identified as such in the Methods section of the paper, with rationale.

Comments from Reviewers:

Reviewer #1: The paper is much improved. I have no further comments. 

Reviewer #2: The main weakness of the study, it means the lack of real novelty of the findings according to the existing literature, has been discussed in the rebuttal by the authors. They are right in claiming that their study represents the most complete and statistically robust evaluation of the EBOV persistence in semen to date. However, I am still convinced that this new study only confirms previous reports and does not represent a breakthrough in the field, even if the analysis of the associations between sociodemographic and acute disease severity and persistence constitutes a new finding. 

The other main criticism, which is the lack of viral isolation data in semen, is still present and the author's answer unfortunately does not allow to lower this issue. Indeed, viral isolation from semen is not feasible in the current study. Instead, they cite a paper from Whitmer (Cell Reports, 2018) to provide data about viral isolation from semen arguing that data from the survivors included in this cohort were presented in this report. However, unless I am mistaken, viral isolation from semen is only presented for 5 survivors hospitalized in the USA in this manuscript. The impossibility to provide data about infectious virus load within semen is unfortunate, as it would have constituted a substantial added value and novelty to the manuscript

Reviewer #3: The authors have implemented most suggested improvements except for the issue of providing validity of their data base for all follow up analyses. 

By only stating the limitations of RT PCR versus virus isolation in the paper and not addressing them will give the reader an insufficient base of their results for the conclusions drawn. PCR ct-values are assay dependent and do not allow a comparison with results from other studies using another PCR assay (also: different assays were applied in the published study). It is understandable, that the test for viable virus in semen can only be achieved with great effort and the described challenges with shipping and laboratory limitations in-country might not be possible to change subsequently. The referencing of the Whitmer paper (2018) is of value but does not allow the reader to get the ct-values of the present study in correlation to virus amounts within the samples. I would like to encourage presenting the data and cut-off rather by means of virus genome copy numbers than by ct-value. Ct-values are only readouts of a machine assay dependent and do not present a functional meaning in contrast to virus genome copy numbers, which is easily obtained by including an internal standard. 

Reviewer #4: I have no additional concerns or edits required for the manuscript from Thorson and colleagues and feel this is an important addition to the literature on Ebola virus persistence.

[LINK]

---

## [Editor Report · Decision Letter 3]

29 Dec 2020

Dear Dr. Thorson,

I am writing concerning your manuscript submitted to PLOS Medicine, entitled “Persistence of Ebola virus in semen among survivors in Sierra Leone: A cohort study of frequency, duration and risk factors.”

We have now completed our final technical checks and have approved your submission for publication. You will shortly receive a letter of formal acceptance from the editor.

Kind regards,

PLOS Medicine